# Delayed Aortic Valve Perforation Caused by Blunt Trauma

**DOI:** 10.3390/diagnostics13030549

**Published:** 2023-02-02

**Authors:** Kazuya Tateishi, Chantal Y. Asselin, Elie M. Elmann, Joseph De Gregorio

**Affiliations:** 1Cardiovascular Services, Englewood Health, Englewood, NJ 07631, USA; 2School of Medicine, University of Limerick, V94 T9PX Limerick, Ireland; 3Cardiac Surgery, Hackensack University Medical Center, Hackensack, NJ 07601, USA

**Keywords:** blunt chest trauma, aortic regurgitation, delayed onset

## Abstract

Traumatic aortic regurgitation (AR) is a rare complication of blunt chest trauma. We described the case of a 35-year-old male who presented to our hospital with shortness of breath 7 years after sustaining blunt chest trauma associated with a motorcycle accident. Transthoracic and transesophageal echocardiogram detected severe AR with two separate jets. The patient was diagnosed with congestive heart failure due to severe AR, and surgical aortic valve replacement was performed. A large perforation of the right coronary cusp likely sustained during the initial blunt chest trauma injury was confirmed surgically. As AR caused by blunt chest trauma can gradually worsen, it is necessary to confirm if there is a history of trauma in patients with severe AR of unknown origin.

**Figure 1 diagnostics-13-00549-f001:**
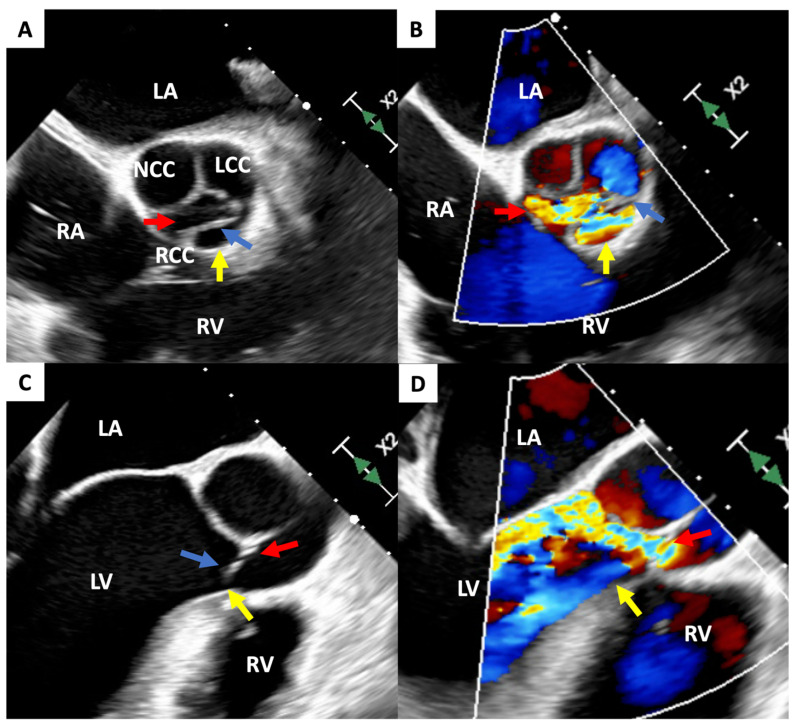
Transesophageal echocardiogram visualizing severe AR. (**A**,**B**): short-axis view, 51 degrees rotation; (**C**): long-axis view, 139 degrees rotation; and (**D**): long-axis view, 133 degrees rotation. Two AR jets are generated. The red arrow indicates regurgitation through the point of incomplete coaptation; the yellow arrow indicates secondary regurgitation through the large perforation in the RCC. The blue arrow indicates the string-like structure of damaged RCC. AR—aortic regurgitation; LA—left atrium; LCC—left coronary cusp; LV—left ventricle; NCC—non-coronary cusp; RA—right atrium; RCC—right coronary cusp; and RV—right ventricle. Traumatic aortic regurgitation (AR) is a rare complication of blunt chest trauma [1]. Although the clinical course of traumatic AR generally follows a rapid progression [2], some patients may have a delayed onset of AR [3]. We report a patient who presented to the hospital in acute decompensated heart failure due to severe AR following a 7-year asymptomatic period after sustaining a blunt chest trauma. Echocardiographic imaging of the heart identified a perforated right coronary cusp (RCC) as the source of AR. This 35-year-old previously healthy male was transferred to our hospital in acute decompensated congestive heart failure. He did not have a history of drug abuse or rheumatic fever. However, he sustained blunt chest trauma 7 years prior due to a motorcycle accident. His injuries, which were treated at the time of the accident, included a fractured cervical spine, a right pneumothorax, and brachial plexopathy. Six days prior to the transfer, he was admitted to the hospital with complaints of sharp back pain and shortness of breath. On admission, he had an elevated blood pressure (159/46 mmHg) with a collapsing pulse, a heart rate of 78bpm, a respiratory rate of 26/minute, and a 95% arterial oxygen saturation. This patient had an obese body habitus (body mass index: 41.61 kg/m^2^) and lower extremity edema. A blood analysis indicated he was not anemic (hemoglobin 13.9 mg/dL), but had elevated levels of brain natriuretic peptide (304 pg/mL), c-reactive protein (1.35 mg/dL), troponin I (103 ng/L), and D-dimer (379 ng/mL). The X-ray and computed tomography scan excluded an aortic dissection but detected mild pulmonary congestion with marked cardiomegaly. An electrocardiogram indicated possible left atrial enlargement and left ventricular (LV) hypertrophy. Upon clinical deterioration, he was transferred to our hospital for further examination and treatment. Transthoracic echocardiogram (TTE) revealed diffuse LV hypokinesis, and a left ventricular ejection fraction (LVEF) of 41.3% with a marked LV dilatation (LV end-systolic diameter of 86 mm). Severe AR was noted on TTE with a pressure half time of 153.5 msec, deceleration slope of 668.0 cm/s^2^, and the visualization of an abnormal jet. Further investigation with a transesophageal echocardiogram (TEE) showed an eccentric jet toward the anterior mitral leaflet through the RCC. Jet width was measured at 6 mm with an effective regurgitant orifice area of 0.4 cm^2^, which suggested a perforated right coronary leaflet (this figure). Coronary angiography and the right heart catheter did not show any significant coronary artery stenosis, but AR was measured as Sellers grade 4 with an elevated pulmonary capillary wedge pressure (27 mmHg).

**Figure 2 diagnostics-13-00549-f002:**
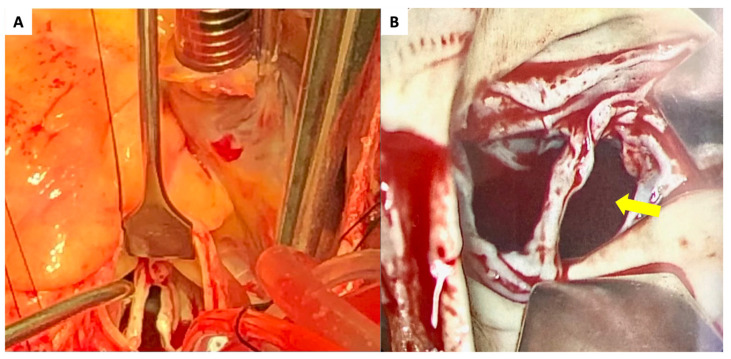
Intraoperative findings of the aortic valve with a wide tear of the RCC. (**A**,**B**) The yellow arrow indicates the large perforation in the RCC. RCC—right coronary cusp. Following these investigations, this patient was definitively diagnosed with congestive heart failure due to severe AR. Surgical aortic valve replacement with a mechanical valve (29 mm St. Jude) was selected as the treatment. A mechanical valve implant was preferred for this patient due to several factors, including the patient’s age, tolerance for long-term anticoagulation, the medical reliability of the valve, and risk factors for re-operation. Intra-operatively, a large perforation of the RCC with a string-like structure originating from the annulus that extended from commissure to commissure was confirmed (this figure). Interestingly, the bases of the other leaflets had thinned and demonstrated stress-related microtears (this figure). At this time, we also repaired the original manubrium-sternum transverse separation, likely caused by the blunt chest trauma sustained during the accident, with plate fixation. Post-operative LVEF did not improve, but the mechanical valve was functioning properly. The patient was discharged on day 9 following transfer to our care without any further adverse events. Aortic regurgitation due to blunt chest trauma is rare, having been reported only 95 times from 1955 to 2015 [4]. This type of injury is typically sustained during horrific traffic accidents or a fall from a height [4]. The mechanism of injury is likely caused by a high transvalvular gradient resulting from increased pressure inside the aortic root against a closed aortic valve and low LV pressure in diastole [5]. The noncoronary cusp (NCC) is most frequently involved in AR due to blunt chest trauma, followed by the RCC [2,4]. The left coronary cusp (LCC) is rarely involved in blunt chest trauma-associated AR [2]. This may be due to two mechanisms: (1) the LCC is a more posterior structure that is less commonly involved in trauma, and (2) the origin of the coronary artery is situated on the LCC, which reduces the pressure sustained by the leaflet [5]. The diagnosis of traumatic AR is traditionally based on 4 criteria, including: (1) a history of blunt chest trauma; (2) absence of a history of heart disease; (3) sudden onset of signs and symptoms of AR; and (4) visualization of severe AR on cardiac imaging such as thoracic aortography or echocardiography [1]. In the present case, this patient met all 4 diagnostic criteria, including the absence of a history of heart disease. A large avulsion of the RCC was confirmed despite a negative history of infective endocarditis or rheumatic fever. Taking this patient’s history and hospital presentation into clinical context, the cause of severe AR must be blunt chest trauma. Notably, symptoms only appeared 7 years after the patient sustained the blunt chest trauma. Given the structural damage of the aortic valve, which included a completely perforated RCC, stress-related microtears and thinning of the NCC and LCC, as well as marked LV dilatation, the formation of regurgitant developed slowly over the course of several years. Typically, the time from the incident to the diagnosis of traumatic AR is within 3 months [2]. However, in some cases, AR from blunt chest trauma can worsen gradually, taking patients up to 10 years or longer to become symptomatic [4]. It is reported that late-presenting patients may originally have had a small, clinically insignificant tear in one of the leaflets that gradually progressed with time as a result of hemodynamic stress [6]. Thus, it is important to confirm if a patient has a history of blunt chest trauma when they present with severe AR of unknown origin. TEE represents the most important tool for evaluating the etiology of regurgitation [7]. In the present case, LV dilatation and severe LV dysfunction were likely caused by chronic, severe AR. Previous studies have reported that LVEF after aortic valve replacement may improve [8]. However, whether LV dysfunction was derived from severe AR or degeneration (i.e., cardiomyopathy) remains unknown. We anticipate improvement in LV function in our patient by optimizing medical therapy following surgical repair of the aortic valve. In conclusion, we treated a patient who developed acute decompensated heart failure due to traumatic AR 7 years following the initial accident. AR due to blunt chest trauma can gradually worsen. Therefore, it is crucial to confirm a history of blunt chest trauma in patients with severe AR of unknown origin.

## Data Availability

All relevant data are presented in this manuscript and further inquiry can be directed to the corresponding author.

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
