# Peer review of "Delayed Aortic Valve Perforation Caused by Blunt Trauma"

_diagnostics, 2023, doi:10.3390/diagnostics13030549_

Round 1
Reviewer 1 Report
SOME COMMENTS:
1) Abstract:
severe AR with abnormal jet.....
Is there any normal jet in patients with aortic regurgitation?
2) Introduction:
have a symptom of acute decompensated heart failure......
Heart failure is a syndrome characterised by a constellation of symptoms.
3) Figure 1:
Red arrow indicates the regurgitation through the point of incomplete coaptation.
The red arrow does not show any regurgitation (no color doppler)
Author Response
Responses to the comments of Reviewer #1
Thank you for the careful and comprehensive review of our manuscript. Our manuscript has been checked by a native English-speaking colleagues. Further, we have modified the manuscript according to their suggestions. Changes we made are written in red in the manuscript.
1) Abstract: severe AR with abnormal jet..... Is there any normal jet in patients with aortic regurgitation?
As the reviewer pointed out, there were 2 separate jets, one of which was caused by perforation of the right coronary cusp (to normal direction), and the other was caused by incomplete coaptation (to abnormal direction). We have changed the Figure 1 as easy to understand for readers, accordingly. We also amended the sentence as follows.
“Transthoracic and transesophageal echocardiogram detected severe AR with 2 separate jets.” (Page 1, Line 10-11)
2) Introduction: have a symptom of acute decompensated heart failure...... Heart failure is a syndrome characterised by a constellation of symptoms.
Thank you for your kind instruction. We amended the sentence as follows.
“We report a patient who presented to hospital in acute decompensated heart failure due to severe AR following a 7-year asymptomatic period after sustaining a blunt chest trauma. Echocardiographic imaging of the heart identified a perforated right coronary cusp (RCC) as the source of AR.” (Page 1, Line 21-24)
3) Figure 1: Red arrow indicates the regurgitation through the point of incomplete coaptation. The red arrow does not show any regurgitation (no color doppler)
We appreciate the reviewer’s comment. The arrows have been provided in the pictures with color doppler. (Figure 1-B and 1-D)
Reviewer 2 Report
Well described rare case, with significant teaching points for aortic specialists. Worth publishing.
Author Response
Responses to the comments of Reviewer #2
Thank you very much for your time and input.
Reviewer 3 Report
This is an interesting case report showing the delayed aortic valve regurgitation due to blunt trauma. My only comment on this report relates to the language writing as there are many grammatical errors. Just list a few:
1. Author list: MD, the qualification is not required.
2. 7 years before- 7 years ago and this should be corrected at other areas throughout the manuscript.
3. Line 20: was suddenly have-suddenly had or suddenly presented with...
4. Line 24, heath male-healthy male.
5. Figure 1: Definition of these abbreviations in the figure (RV, LA, LV, RCC, LCC etc) should be provided in the figure legend.
Author Response
Responses to the comments of Reviewer #3
Thank you for the careful review of our manuscript. We are sorry for our grammatical errors. Our manuscript has been checked by a native English-speaking colleagues. Further, we have modified the manuscript according to their suggestions. Changes we made are written in red in the manuscript.
1) Author list: MD, the qualification is not required.
Thank you for pointing out. We deleted the qualification from the Author lists.
2) 7 years before- 7 years ago and this should be corrected at other areas throughout the manuscript.
We amended the sentences as follows.
“We described a case of a 35-year-old male who presented to our hospital with shortness of breath 7 years after sustaining blunt chest trauma associated with a motorcycle accident.” (Page 1, Line 8-10)
“However, he sustained a blunt chest trauma 7 years prior due to a motorcycle accident.” (Page 1, Line 28-29)
“6 days prior to the transfer, he was admitted to hospital with complaints of sharp back pain and shortness of breath.” (Page 1, Line 31-32)
3) Line 20: was suddenly have-suddenly had or suddenly presented with...
We amended the sentence as follows.
“We report a patient who presented to hospital in acute decompensated heart failure due to severe AR following a 7-year asymptomatic period after sustaining a blunt chest trauma.” (Page 1, Line 21-23)
4) Line 24, heath male-healthy male.
We changed the word “heath” to “healthy”. (Page 1, Line 26)
5) Figure 1: Definition of these abbreviations in the figure (RV, LA, LV, RCC, LCC etc) should be provided in the figure legend.
We are sorry for lack of abbreviations. We added these abbreviations in the figure legend. (Page 2, Line 58-60)
Round 2
Reviewer 1 Report
6 days prior to the transfer, he was admitted to hospital with complaints
Six days..............